# Hybrid Concept-based Models: Using Concepts to Improve Neural Networks' Accuracy

Tobias A. Opsahl[1]

[1]University of Oslo
`tobiasao@uio.no`

## Abstract

Most datasets used for supervised machine learning consist of a single label per data point. However, in cases where more information than just the class label is available, would it be possible to train models more efficiently? We introduce two novel model architectures, which we call *hybrid concept-based models*, that train using both class labels and additional information in the dataset referred to as *concepts*. In order to thoroughly assess their performance, we introduce *ConceptShapes*, an open and flexible class of datasets with concept labels. We show that the hybrid concept-based models can outperform standard computer vision models and previously proposed concept-based models with respect to accuracy. We also introduce an algorithm for performing *adversarial concept attacks*, where an image is perturbed in a way that does not change a concept-based model's concept predictions, but changes the class prediction. The existence of such adversarial examples raises questions about the interpretable qualities promised by concept-based models.

## 1 Introduction

Understanding model behaviour is a crucial challenge in deep learning and artificial intelligence [1–4]. Deep learning models are inherently chaotic, and give little to no insight into why a prediction was made. In computer vision, early attempts of explaining a model's prediction consisted of assigning pixel-wise feature importance, referred to as *saliency maps* [5–8]. Despite gaining popularity and being visually appealing, a large number of experiments show that saliency maps perform a poor job at actually explaining model behaviour [2, 3, 9–13].

Recently, several *concept-based models* have been proposed as inherently interpretable [14–18]. These models are restricted to perform the downstream prediction only based on whether it thinks some predefined *concepts* are present in the input or not, where concepts are defined as human meaningful features. This way, the downstream predictions can be interpreted by which concepts the model thought were in the data.

However, recent experiments have highlighted is-

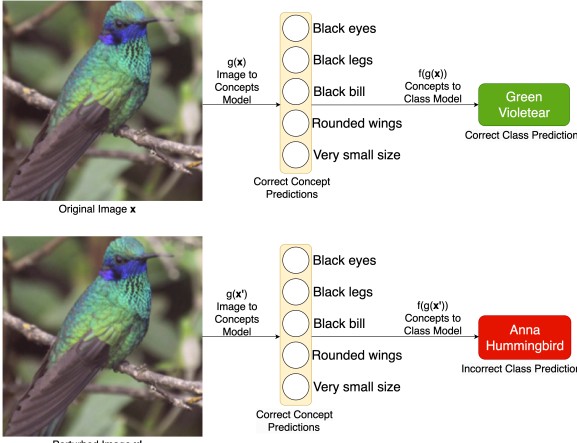

**Figure 1. Adversarial Concept Attack.** Images are perturbed in a way that does not change a concept-based model's concept predictions, but change the class prediction. This brings into question the interpretable qualities of these models.

sues with concept-based models' interpretability. This is mainly due to the concept predictions encoding more information than just the concepts, referred to as *concept leakage* [19, 20]. We further add evidence to the lack of interpretability in concept-based models by introducing *adversarial concept attacks* (see Figure 1).

Due to the evidence demonstrating the limitations of interpretability in concept-based models, we will shift our focus away from interpretability and instead use the framework of concept-based models to improve the performance of the models. We present two new model architectures aimed to achieve this.

Our proposed model architectures use both concept predictions and information not interfering with the concepts to make the downstream prediction. This way, the models can use the concept predictions if they are helpful for the downstream task, but can also rely on a skip connection to encode information about the data not present in the concepts. We propose these models to better utilize the available information in datasets with concepts.

A challenge in this research area is that the most popular datasets used for benchmarking concept-based models have shortcomings that we argue make them unsuitable to use as benchmarks, and

Proceedings of the 26th Northern Lights Deep Learning Conference (NLDL), PMLR 307, 2026.

we therefore developed a new class of flexible concept datasets. The Caltech-UCSD Birds-200-2011 dataset (CUB) [21] is the most widely used concept dataset, where the downstream task is to classify images among 200 classes of bird species, and is widely used to benchmark concept-based models [14–18, 22–24]. Despite its popularity, there are various problems with the concept labelling, which was done by non-experts. Therefore, the dataset is processed with a class-wise majority vote of the concept labels, so every class has the exact same concept labels [15]. Unfortunately, not only has this led to mistakes where instances of a class have different concepts [18], but ambiguity of concepts is very common. For instance, there are images of birds where one can not see its tail, belly or wings properly, but it may still be labelled with concepts relating to those body parts [18].

Another popular concept dataset is *Osteoarthritis Initiative* (OAI) [15, 17, 24, 25]. The main problem with this dataset is the lack of availability. Since it uses medical data, access to the dataset needs to be requested, and the processed version is not directly available. Moreover, the computational resources used to process it were "*several terabytes of RAM and hundreds of cores*" [26], which is not available for many researchers.

Our contributions are as follows:

- **Novel model architectures:** We propose novel model architectures which we call *hybrid concept-based models*. Unlike previously proposed concept-based models, ours are motivated by performance, not interpretability. We conduct experiments that show that they can outperform other computer vision and concept-based models.

- **New concept datasets:** In order to properly assess the performance of concept-based models, we propose a new set of openly available datasets with concepts called *ConceptShapes*.

- **Adversarial concept attacks:** We propose an algorithm for generating adversarial examples specifically for concept-based models. These examples further question concept-based models' interpretable qualities.

## 2 Related Work

### 2.1 Concept-based Models

Concept-based models first predict some predefined concepts in the dataset, then use those concept predictions to predict the downstream task. This way, the final prediction can be interpreted by which concepts the model thought were present in the input. One of the first and most popular concept-based

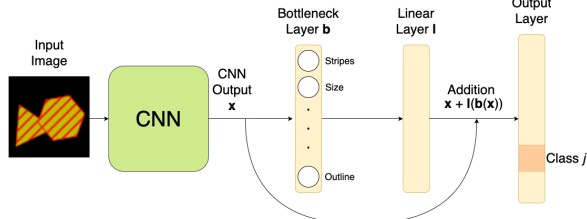

**Figure 2. Architecture of a CBM-Res** used for computer vision. The model uses both concept predictions and a skip connection that hops over the bottleneck layer to perform the output prediction. The architecture can be adapted to a CBM-Skip by performing concatenation instead of addition before the output layer.

models is the *concept bottleneck model* (CBM) [15], which is a neural network with a *bottleneck layer* that predicts the concepts. The model is trained both using the concept labels and the target labels.

Several alternatives to the CBM architecture have been proposed. *Concept-based model extraction* (CME) [14] may use a different hidden layer for the various concepts. *Post-hoc concept bottleneck models* (PCBM) [23] first learn the concept activation vectors (CAVs) [27] of the concepts, and then project embeddings down on a space constructed by CAVs. *Concept embedding models* (CEM) [22] produces vectors in a latent space of concepts that are different for presence and absence of a concept and predicts the probabilities of concepts being present.

Several experiments show that the concept predictions encode more information than just the concepts, and therefore that they are unsuitable to use as interpretation of the models' behaviour [19, 20]. It has also been shown that concept-based models are susceptible to adversarial attacks that change the concept predictions, but not the class predictions [24].

## 3 Methods

### 3.1 New Model Architectures

We propose two novel model architectures. The first is based on a CBM [15], but uses an additional skip connection that does not go through the concept bottleneck layer (see Figure 2). The skip connection can be implemented either as a residual connection [28] or a concatenation [29], and we refer to the models as *CBM-Res* and *CBM-Skip*, respectively. This way, the model can use both the concept prediction and information not interfering with the concepts to make the final downstream prediction.

The other proposed architecture predicts the concepts sequentially throughout the neural network's layers, instead of all at once (see Figure 3). All of the concept predictions are concatenated together, along with the final hidden layer, and given as input

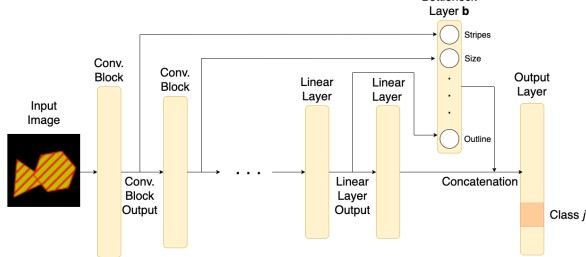

**Figure 3. Architecture of a Sequential Concept Model (SCM)**. The concepts are predicted sequentially throughout the layers, and concatenated together with the final hidden layer before the output layer, which produces the downstream predictions.

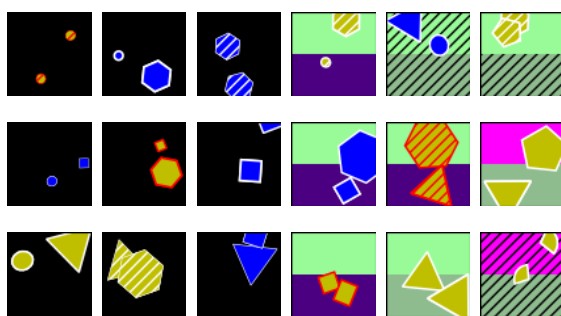

**Figure 4.** Images from different classes of two ConceptShapes datasets. **Left:** Nine different images from a 10-class 5-concept dataset. **Right:** Nine different images from a 21-class 9-concept dataset.

to the output layer. We refer to this model as the *Sequential Concept Model* (SCM).

We use a loss function constructed by a weighted sum of a concept loss and a task loss, similarly to the *joint bottleneck* proposed for the CBM [15].

All of the proposed models are compatible with transfer learning, where the first part of the model can be a large pre-trained network. We present the models in the domain of classification, but they can easily be adapted to regression by replacing the output layer with a single node.

## 3.2 Introducing ConceptShapes

To accurately assess the performance of concept-based models, we have developed a class of flexible synthetic concept datasets called *ConceptShapes*. The input images consist of two shapes, where the position and orientation are random, and the downstream task is to classify which combination of shapes that are present (see Figure 4). Some examples of target classes are "triangle-rectangle", "triangle-triangle" and "hexagon-pentagon". Depending on how many shapes that are used, the dataset contains 10, 15 or 21 classes.

The key feature of the datasets is that various binary concepts are present, such as the colour of the shapes, outlines and background. Given a class, some predefined concepts are drawn with a high probability $s \in [0.5, 1]$, and the others with a low probability $1 - s$. The hyperparameter $s$ can be chosen by the user. When $s = 0.5$, the concepts are drawn independently of the classes, and when $s = 1$, the concepts are deterministic given the class. The datasets can be created with either five or nine concepts.

The datasets are flexible with regards to the relationship between the concepts and the classes, and the number of concepts, classes and data. This way, the difficulty of correctly classifying the images can be tuned by the amount of classes and the amount of data, and the information in the concepts can be tuned by the amount of concepts and the

value of $s$. Further details about the dataset can be found in Appendix B. The code for generating the ConceptShapes datasets can be found at `https://github.com/Tobias-Opsahl/ConceptShapes`.

Although synthetic datasets may have less complex patterns than datasets with real images, there are also clear benefits of using them. They are precisely labelled, there is no ambiguity in the concepts and they have many flexible parameters. Therefore, we believe that ConceptShapes can provide a useful addition to the existing benchmark datasets for concept-based models.

## 3.3 Adversarial Concept Attacks

We propose an algorithm for producing *adversarial concept attacks*, which given a concept-based model and input images, produces identically looking images that give the same concept predictions, but different output predictions. The algorithm is based on *projected gradient descent* (PGD) [30], which iteratively updates an input in the direction which maximizes the classification error of the model, and projects the alteration on an L-infinity ball around the original input. In each iteration, we add a step where we check if the alteration causes the model to almost change the predictions of the concepts. If so, we check which pixels that are altered in a direction that changes the concept predictions, and multiply those pixels' alterations by a number in $[-1, 0]$. The complete algorithm is covered in Appendix C.

Our approach differs from the adversarial attacks for concept-based models done by Sinha et al. [24], which altered the concept predictions, and not the class predictions.

## 4 Experiments

We show that the hybrid concept-based models achieve the highest test set accuracy on multiple datasets. In order to examine how the models perform with different amounts of data, we train and test the models on various smaller subsets of the

datasets. We also investigate how well the concepts are learned.

## 4.1 Datasets

### 4.1.1 Caltech-UCSD Birds-200-2011 (CUB)

The CUB dataset [21] consists of $N = 11,788$ images of birds, where the target is labelled among 200 bird species. The original dataset contains 28 categorical concepts, which makes 312 binary concepts when one-hot-encoded. The processed version used for benchmarking concept-based models [15] removed sparse concepts and used a majority vote on the concept labels, so that every class has the exact same concepts, ending up with 112 binary concepts. The dataset is split in a 50%-50% training and test split, and we use 20% of the training images for validation. We train and evaluate on six different subset sizes.

### 4.1.2 ConceptShapes

We experiment with many different ConceptShapes configurations. First, we set the probability $s$ to be 0.98, in order to make the concepts useful, but not deterministic given the class, and experiment with different amounts of classes. Additionally, we use different values of $s$ to explore how the correlation between the concepts and the classes influence the models' performance. We explore even more configurations of ConceptShapes in Appendix A.2.

The datasets are generated with 1000 images in each class, and we split them into 50%-30%-20% train-validation-test sets. We train and evaluate on subsets sizes with 50, 100, 150, 200 and 250 images in each class, drawn from the 1000 images created.

## 4.2 Setup

We compare our proposed models against a CBM, which we refer to as *vanilla CBM*, and a convolutional neural network (CNN) [31] not using the concepts at all, referred to as the *standard model*. Additionally, we also include an *oracle model*, which is a logistic regression model trained only on the true concept labels, not using the input images. We call it an *oracle* since it uses true concept labels at test time, which are usually unknown, and do this in order to measure how much information there is in the concepts alone.

The models' accuracies are evaluated on a held-out test set, which is the same for every subset configuration. We perform a hyperparameter search for each model and each subset configuration using the validation sets. The details of the hyperparameter settings are covered in Appendix D.

The models trained on CUB use a pre-trained and frozen ResNet-18 [28] as the convolutional part

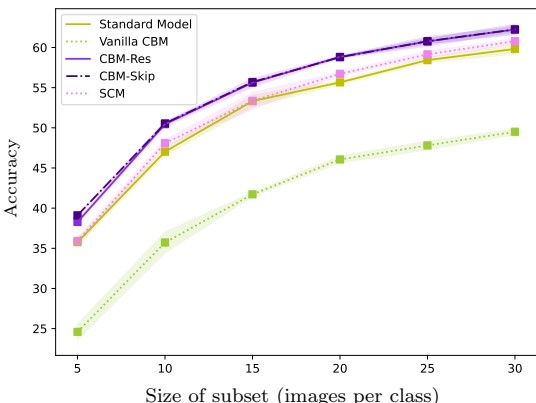

**Figure 5. Test set accuracies on the CUB dataset.** The x-axis indicates the average amount of images included in the training and validation dataset for each class, where the rightmost point corresponds to the full dataset. The results are averaged over three runs and include 95% confidence intervals. The oracle model consistently got 100% accuracy and is omitted.

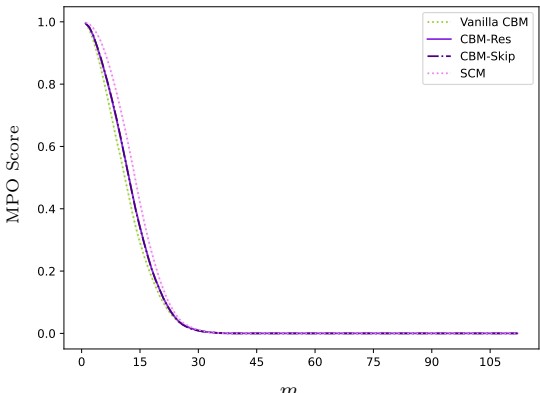

**Figure 6. MPO scores on the CUB dataset.** The y-axis indicates the proportion of images with $m$ or more concept prediction mistakes. The results are averaged over three runs and include 95% confidence intervals. We used the full dataset.

of the model, while the models trained on ConceptShapes are trained from scratch with a smaller model. The details about the setup are explained in Appendix E and the code for running the experiments can be found at https://github.com/Tobias-Opsahl/Hybrid-Concept-based-Models.

We also record the *Misprediction Overlap* (MPO) [14] to measure the quality of the concept predictions. The MPO measures the ratio of images that had $m$ or more concept mispredictions. We use $m = 1, 2, \ldots, k$, where $k$ is the amount of concepts in the dataset.

We run a grid search to find the most successful adversarial concept attack ratio, meaning the ratio of images that change the model's class prediction, but not the concept predictions. We use the best vanilla CBM found in the hyperparameter searches and compare the results to PGD [30].

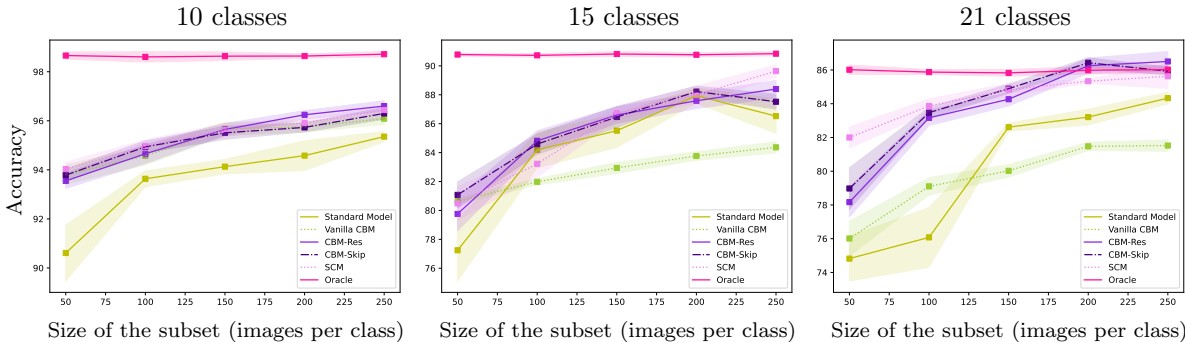

**Figure 7. Test set accuracy on ConceptShapes** with nine concepts and $s = 0.98$. The plots show that the hybrid concept-based models perform better than the benchmark models. The x-axis denotes how many training and validation images that were included in each class. The metrics are averaged over ten runs and include 95% confidence intervals.

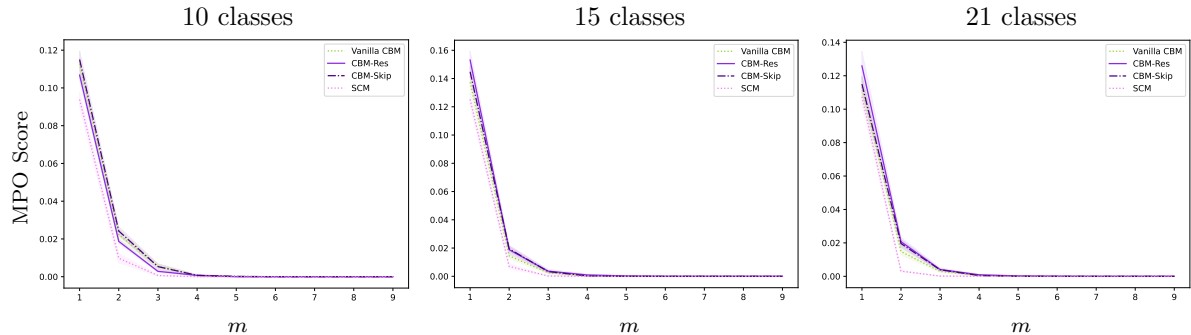

**Figure 8. MPO scores on ConceptShapes.** All three plots shows that the concepts are properly learned by all models, where about 85% of the images have no concept mispredictions, and only about 2% have more than two. We used 250 training and validation images in each class. The metrics are averaged over ten runs and include 95% confidence intervals.

## 5 Results and Discussion

### 5.1 Improved Accuracy

The test set accuracies can be found in Figure 5, 7 and 9. We observe that the hybrid concept-based models generally have the best performance. When $s = 0.5$ (Figure 9), the concepts provide no additional information that helps predict the classes. However, the hybrid concept-based models do not perform worse than the CNN, suggesting they are able to assign low weights to the bottleneck layer when the concepts are irrelevant. When $s$ increases, the performance of all models does as well, and the gap between the hybrid concept-based models and the benchmark models becomes larger.

The oracle model, using only the true concept labels at test time, serves as a baseline for how much information there is in the concepts alone. When $s = 0.5$ (Figure 9), the oracle model has a test set accuracy of about 10%, similarly to random guessing. The oracle model's accuracy increases when $s$ increases (Figure 9), but decreases when there are more classes (Figure 7). Since $s$ controls the information in the concepts, and an increase in the number of classes also increases the difficulty of

classifying, this behaviour is expected. With a low $s$ or a large number of classes, the hybrid concept-based models perform better than the oracle model (see Figure 7, 9).

### 5.2 CUB Concepts are not Learned

When inspecting the MPO plots for CUB in Figure 6, we see that none of the concept-based models learn the concepts properly. About 50% of the images are predicted with 15 or more mistakes in the concept predictions for all models, which is about as good as random guessing, since the labels are one-hot-encoded and sparse. This observation is consistent with earlier work that has pointed out that the concepts in CUB are ambiguous and sometimes wrong [18].

When inspecting the MPO plot for ConceptShapes in Figure 8 and 10, we do however see that the concepts are properly learned, even when $s = 0.5$ and the concepts are useless for predicting the class. This suggests that the concepts in ConceptShapes are not ambiguous, and the datasets therefore serve as better benchmark datasets for concept-based models. The SCM predicts the concepts better than the other

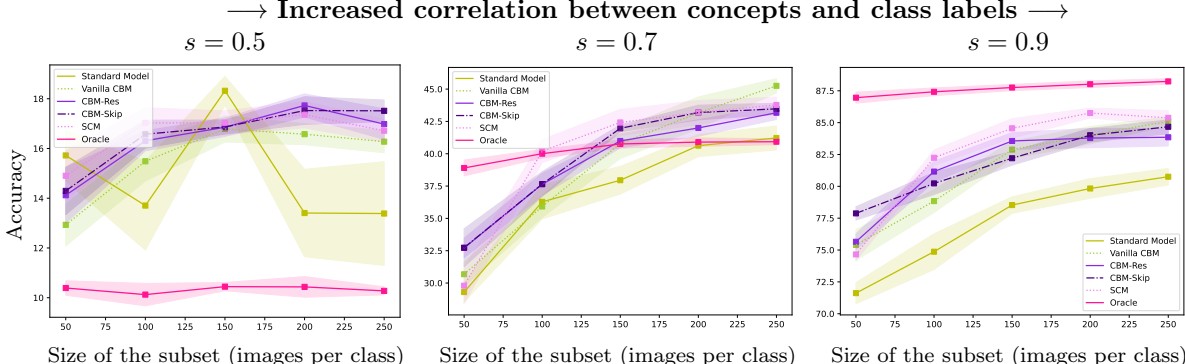

**Figure 9. Test set accuracy on ConceptShapes** with different values of $s$, using ten classes and nine concepts. Higher values of $s$ means more correlation between the concepts and the classes.

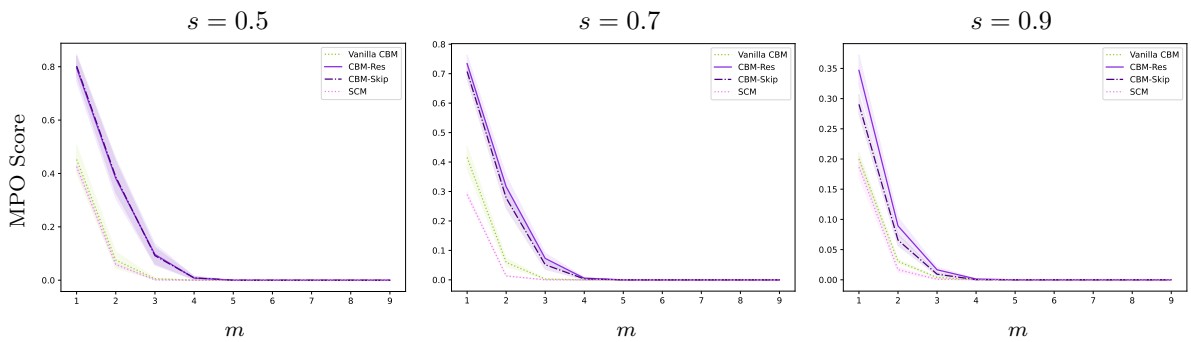

**Figure 10. MPO scores on ConceptShapes** with different values of $s$ using ten classes and nine concepts. The majority of images are have less than two concept mispredictions, even when the concepts are irrelevant for the classes ($s = 0.5$). The SCM performs better than the other models.

|  | Adversarial Concept Attack Success Rate | PGD Success Rate |
|---|---|---|
| **CUB** **112 concepts** | 57.4% | 16.2% |
| **ConceptShapes with** **10 classes and 5 concepts** | 35.5 % | 31.4% |
| **ConceptShapes with** **21 classes and 9 concepts** | 26.6% | 22.5% |

**Table 1.** Success rate of adversarial concept attacks on images in the test sets. An attack is considered a success when the class prediction is changed, but not the concept predictions.

concept-based models, which might be due to having a higher number of parameters directly connected to the concept predictions.

## 5.3 Adversarial Concept Attacks

The results of the adversarial concept attacks can be found in Table 1. We see that a substantial amount of images are perturbed with success, and the algorithm is more effective than PGD.

Since the concept predictions are used as the interpretation of the model behaviour, but the same interpretation can lead to vastly different model behaviour, we suggest that this experiment questions the interpretable qualities of concept-based models.

## 6 Conclusions

We proposed new hybrid concept-based models motivated by improving performance, and demonstrated their effectiveness on CUB and several Concept-Shapes datasets. The proposed models train using both the class label and additional concept labels. In all of the datasets we experimented with, the hybrid concept-based models performed better than previously proposed concept-based models and the standard computer vision models.

We also introduced ConceptShapes, a flexible class of synthetic datasets for benchmarking concept-based models. Finally, we demonstrated that concept-based models are susceptible to adversarial concept attacks, which we suggest are problematic for their promised interpretable qualities.

In future work, we would like to apply hybrid concept-based models in the domain of reinforcement learning, where concepts such as agent rotation, position and velocity can be automatically calculated, and do not need manual labelling.

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

# A    More Experiments

## A.1    Hard Bottleneck

We also conducted experiments where we rounded off the concept predictions to binary values in the concept-based models, referred to as a *hard bottleneck*. The results can be seen in Figure A.1 and Figure A.2. We see that the vanilla CBM's performance is reduced, while the hybrid concept-based models are not changed much. Comparing the MPO plots (Figure A.2 and Figure 6) shows little change in how well the concepts were learned.

## A.2    ConceptShapes Experiments

We experiment with using five concepts instead of nine, with the results plotted in Figure A.3. We see that the oracle model has a lower accuracy, indicating there is less information in the concepts alone. Thus, the gap between the standard model and the hybrid concept-based models is a little narrower, although the hybrid concept-based models still perform the best in general. When inspecting the MPO plot in Figure A.4, we see that the concepts are still learned by all the models.

# B    ConceptShapes Details

We now explain the ConceptShapes datasets in greater detail. The crucial feature of the datasets are the concepts. All of them are binary and independent, meaning any combination of concepts is possible. The five first concepts are based on the two shapes in the image, while the last four optional concepts are based on the background. We now describe the concepts one-by-one, and visualizations are available in Table B.1 and Table B.2. We start with the five concepts that influence the shapes:

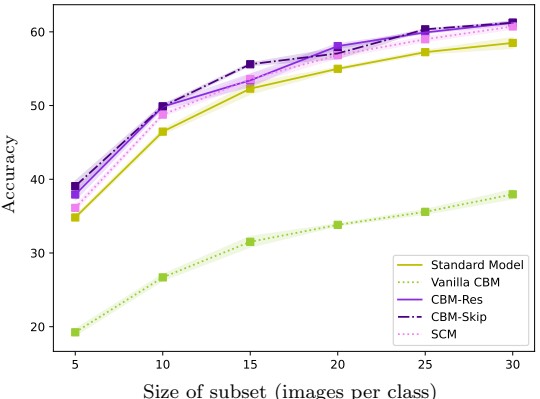

**Figure A.1. Test set accuracies on the CUB dataset with a hard bottleneck.** The vanilla CBM suffers from substantially lower performance, while the hybrid concept-based models' performances are not changed much. The metrics are averaged over three runs and contains 95% confidence intervals.

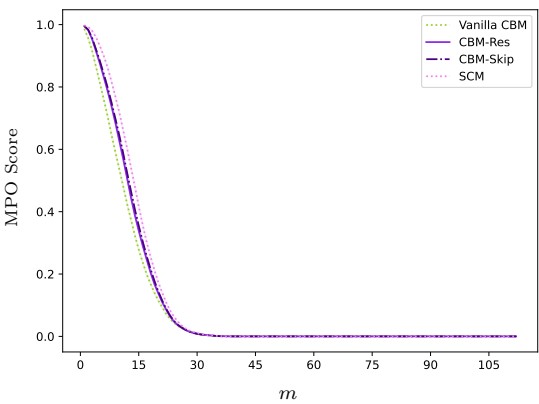

**Figure A.2. MPO scores on the CUB dataset with a hard bottleneck.** There is little to no change in how well the concepts are predicted when using a hard bottleneck. We used the full dataset.

1. **Big shapes**. Every shape had two intervals of sizes to be randomly drawn from. One interval corresponded to the small figures, and the other to big ones.

2. **Thick outlines**. The outlines of the shapes were drawn from one of two intervals. One corresponded to a thin outline, and the other to a thick one.

3. **Face colour**. There were two possible colours for the shapes, blue and yellow.

4. **Outline colour**. The shapes had two possible outline colours, red and white.

5. **Stripes**. Some shapes were made with stripes, and some were not. The stripes were in the same colour as the outline.

All of the concepts apply to the whole image. For

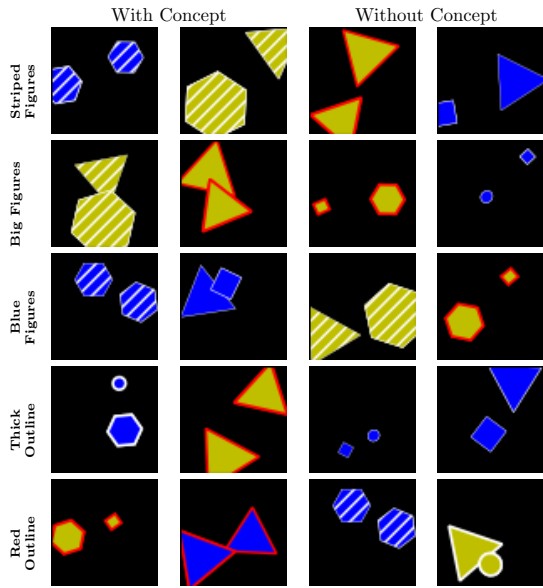

**Table B.1. Overview of the five concepts regarding the shapes.** Each row corresponds to one concept. The two leftmost columns of images have the concept, and the two rightmost columns do not have the concept. All of the images are from a 5-concept dataset, hence the black background. These five concepts are also present in the 9-concept datasets.

instance, if the image gets the thick outline concept, both shapes in the image get a thick outline.

The datasets that use nine concepts have all five of the concepts above, in addition to four more. While all of the five-concept datasets have black backgrounds, the nine-concept datasets split the background in two and use the colour and stripes of the background as additional concepts.

6. **Upper background colour**. The upper-half of the background would either be magenta or pale-green.

7. **Lower background colour**. The lower-half of the background would be either indigo or dark-sea-green.

8. **Upper background stripes**. This represented whether there were black stripes present in the upper background or not.

9. **Lower background stripes**. This represented whether there were black stripes present in the lower background or not.

To summarize, some of the image's visuals are determined by the concepts, some by the classes and some by randomness. The two shapes (from triangle, square, pentagon, hexagon, circle and wedge) are determined by the class. The shapes' size, colour and outline are determined by the concepts. If the dataset uses nine concepts, the background colour and stripes are also determined by the concepts.

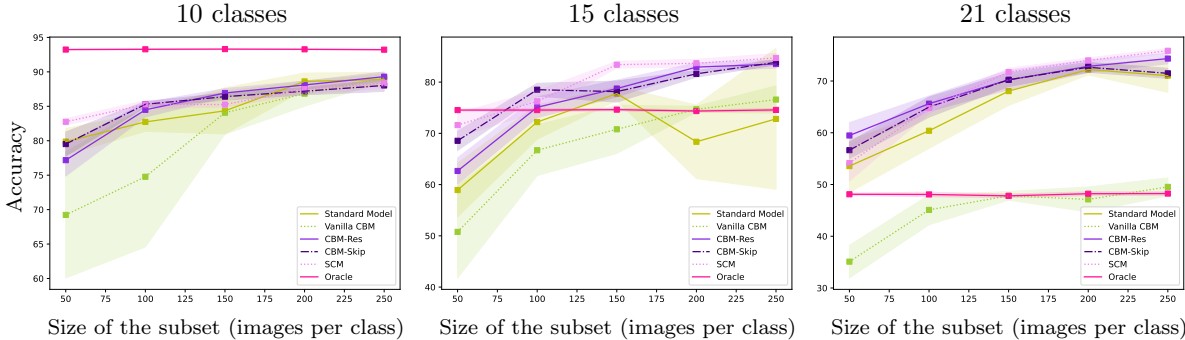

**Figure A.3. Test set accuracy on ConceptShapes** with five concepts and $s = 0.98$. The metrics are averaged over ten runs and include 95% confidence intervals.

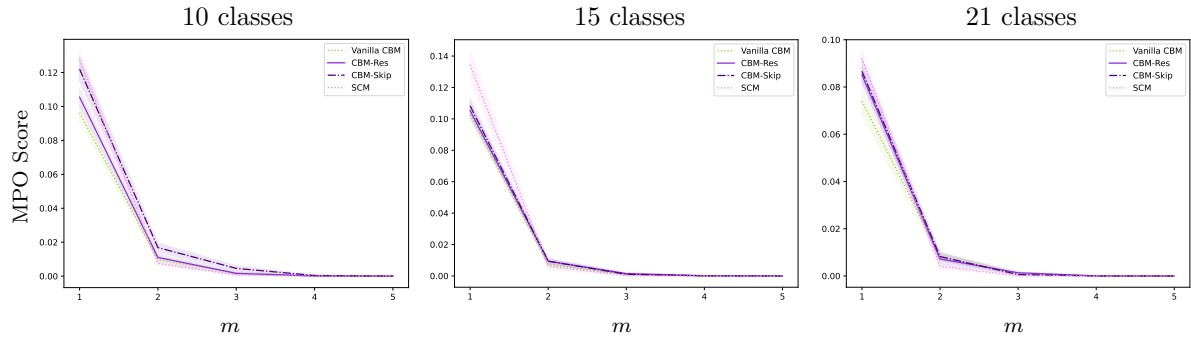

**Figure A.4. MPO scores on ConceptShapes** with $s = 0.98$. We used 250 images in each class. The metrics are averaged over ten runs and include 95% confidence intervals.

The shapes' position and rotation are determined randomly, regardless of which class or concepts they have.

# C   Adversarial Concept Attacks

We describe a high level overview of the algorithm for performing adversarial concept attacks, while the full algorithm can be found in Algorithm 1. We do PGD [30] with an additional step. For each concept, if the concept prediction for the perturbed image is close to changing compared to the original image, we consider the concept as *sensitive*. We control this with a *sensitivity threshold* $\gamma \in [0, \infty)$, so that a concept is considered sensitive if its logits are in the interval $[-\gamma, \gamma]$. We get an initial perturbation in each iteration from following the gradient of the loss with respect to the pixels of the images, similar to PGD. The perturbation is multiplied with a mask $M$, which is constructed so that it is 1 for pixels that do not influence sensitive concepts to be even closer to a change of prediction, and $\beta \in [-1, 0]$ for pixels that do. We loop over the sensitive concepts and iteratively update $M$, and use the notation $\mathbb{I}_\beta(A = B)$ to denote an elementwise indicator function, so that it is 1 if elements in the same place in $A$ and $B$ are equal, and $\beta$ if not. The Hadamard product

$\odot$ represents elementwise multiplication.

The algorithm might terminate with failure due to several reasons. In any steps, if the concept predictions are changed, we terminate. If the mask $M$ has only $\beta$ as elements, the perturbation will not go in a direction that changes the class, and we therefore stop. Finally, if the maximum amount of steps are taken, we also terminate.

The algorithm can be improved in many ways, but our intention is to demonstrate that such adversarial examples are possible and easy to generate, not to make the best algorithm to do so. One possible improvement might for instance be to be able to backtrack if the concept predictions are changed.

When tuning hyperparameters for the adversarial concept attacks, we chose to tune the step size $\alpha$, which is multiplied with the perturbation in each step, and the sensitivity threshold $\gamma$, which determines how close a concept prediction needs to be to change before we try to cancel out its changes.

For the ConceptShapes datasets, we used a grid search with step size $\alpha \in \{0.003, 0.001, 0.00075\}$ and sensitivity threshold $\gamma \in \{0.1, 0.05, 0.01\}$. For CUB, the values were $\alpha \in \{0.0001, 0.000075, 0.00005\}$ and $\gamma \in \{0.1, 0.075, 0.05, 0.02\}$. These values were chosen after some initial experimentation. The best values were $\alpha = 0.001, \gamma = 0.1$ and $\alpha = 0.000075, \gamma = 0.1$, respectively. In order to reduce the running time, we sampled 200 images from the training set

**Algorithm 1:** Adversarial Concept Attack Algorithm

---

**Result:** Perturbed image $\widetilde{\mathbf{x}}$ of $\mathbf{x}$, such that CMB $h$ misclassifies $\widetilde{\mathbf{x}}$, but the concept predictions are the same for $\widetilde{\mathbf{x}}$ and $\mathbf{x}$, or 0 for a failed run.

**Input:** Input image $\mathbf{x} \in \mathbb{R}^d$.

Class label $y \in [1, \ldots, p]$.

CBM $h : \mathbb{R}^d \to \mathbb{R}^p$ with input-to-concept function $g : \mathbb{R}^d \to \mathbb{R}^k$, such that $\operatorname{argmax}(h(\mathbf{x})) = y$.

Sensitivity threshold $\gamma \in [0, \infty)$.

Step size $\alpha \in (0, 1)$.

Deviation threshold $\epsilon \in \mathbb{R}^d$.

Max iterations $t_{\max} \in \mathbb{N}_1$.

Gradient weight $\beta \in [-1, 0]$.

Valid pixel range $[x_{\min}, x_{\max}]$.

$\widetilde{\mathbf{x}}_0 \leftarrow \mathbf{x}$ // Adversarial example

$\hat{\mathbf{c}} = g(\mathbf{x})$ // Original concept logits

$\hat{\mathbf{c}}_b = \mathbb{I}(\sigma(\hat{\mathbf{c}}) > 0.5)$ // Original binary predictions

**for** $t = 0$ **to** $t_{max}$ **do**

    $\widetilde{\mathbf{c}} \leftarrow g(\widetilde{\mathbf{x}}_t)$

    $\widetilde{\mathbf{c}}_b = \mathbb{I}(\sigma(\widetilde{\mathbf{c}}) > 0.5)$ // New concept predictions

    **if** $\widetilde{\mathbf{c}}_b \neq \hat{\mathbf{c}}_b$ **then**

        **return** 0 // Changed concept predictions

    **if** $\operatorname{argmax}(h(\widetilde{\mathbf{x}}_t)) \neq y$ **then**

        **return** $\widetilde{\mathbf{x}}_t$ // Success

    $\hat{\mathbf{p}}_t = \operatorname{sign}(\nabla_{\widetilde{\mathbf{x}}_t} L(h(\widetilde{\mathbf{x}}_t), \mathbf{y}))$ // Initial perturbation

    Initialize $M_t \in \mathbb{R}^d$ with all elements as ones

    **for** $j = 0$ **to** $k$ **do**

        **if** $\widetilde{c}_j$ *in* $[-\gamma, \gamma]$ **then**

            $\mathbf{q}_j = \operatorname{sign}(\nabla_{\widetilde{\mathbf{x}}_t} g(\widetilde{\mathbf{x}}_t)_j)$

            $M_{t,j} \leftarrow \mathbb{I}_\beta(\hat{\mathbf{p}}_t \odot \mathbf{q}_j \neq \operatorname{sign}(g(\mathbf{x})_j))$

            $M_t \leftarrow \min(M_t, M_{t,j})$

    **if** *All entries in $M_t$ equal $\beta$* **then**

        **return** 0 // All $\beta$ mask

    $\mathbf{p}_t = \hat{\mathbf{p}}_t \odot M_t$ // Final perturbation

    $\widetilde{\mathbf{x}}' = \Pi_{[\mathbf{x}-\epsilon, \mathbf{x}+\epsilon]}(\widetilde{\mathbf{x}}_t + \alpha \mathbf{p}_t)$ // Projection

    $\widetilde{\mathbf{x}}_{t+1} = \operatorname{clamp}(\widetilde{\mathbf{x}}', x_{\min}, x_{\max})$

**return** 0 // Max iterations exceeded

---

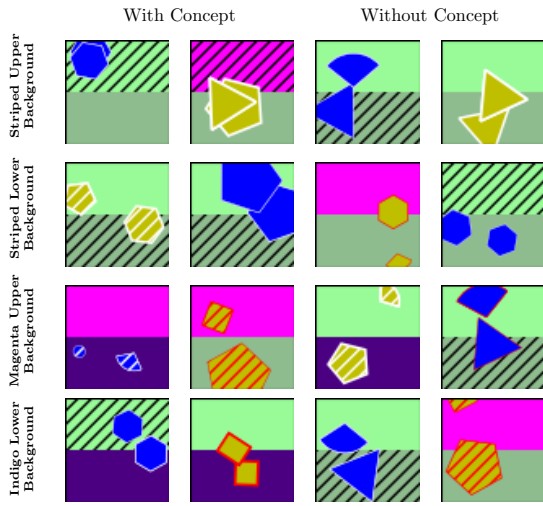

**Table B.2. Overview of the four concepts that relate to the background.** These four concepts are only present in the 9-concept datasets. Each row corresponds to one concept. The two leftmost columns of images have the concept, while the two rightmost columns have not.

that were correctly predicted. We used 800 max steps for ConceptShapes and 300 for CUB.

We ran the grid search with $\beta = -0.3$, which is the weight multiplied with pixels that would make concept predictions change, and $\epsilon = 1$, which determines where the adversarial images get projected back on. After the grid search, we performed a line search on $\beta \in \{0.1, 0, -0.1, -0.3, -0.5, -0.7, -1\}$. The results were very similar, but slightly better for $\beta = -0.1$. The success rates were calculated using all of the images in the test set where the model originally predicted the correct class.

## D Hyperparameter Details

For the ConceptShapes datasets, the learning rates were sampled from values in $\{0.05, 0.01, 0.005, 0.001\}$, and dropout probabilities from $\{0, 0.2, 0.4\}$. The standard model also searched for an exponential decay parameter to the linear learning scheduler in $\{0.1, 0.5, 0.7, 1\}$, applied every five epochs. The concept-based models set the decay parameter to 0.7 and searched for a weight balancing the concept loss function and the class loss function. They were $\{(100, 0.8), (100, 0.9), (5, 1), (10, 1)\}$, where the first element in the tuples represents the weight multiplied with the concept loss, and the second is an exponential decay parameter, applied to the weight every epoch. All of the models had an equal amount of hyperparameter trials.

For the CUB dataset, we set the dropout probability to 0.15 and searched for learning rates in $\{0.001, 0.0005, 0.0001\}$. The standard model

| Model | Total Parameters | Trainable Parameters | Frozen Parameters |
|---|---|---|---|
| ConceptShapes models with 10 classes and 5 concepts | | | |
| Standard model | 139 578 | 139 578 | 0 |
| Vanilla CBM | 137 583 | 137 583 | 0 |
| CBM-Res | 138 527 | 138 527 | 0 |
| CBM-Skip | 138 399 | 138 399 | 0 |
| SCM | 152 849 | 152 849 | 0 |
| ConceptShapes models with 21 classes and 9 concepts | | | |
| Standard model | 139 941 | 139 941 | 0 |
| Vanilla CBM | 138 053 | 138 053 | 0 |
| CBM-Res | 139 758 | 139 758 | 0 |
| CBM-Skip | 139 614 | 139 614 | 0 |
| SCM | 167 579 | 167 579 | 0 |
| CUB models | | | |
| Standard model | 11 425 032 | 248 520 | 11 176 512 |
| Vanilla CBM | 11 371 880 | 195 368 | 11 176 512 |
| CBM-Res | 11 416 952 | 240 440 | 11 176 512 |
| CBM-Skip | 11 428 088 | 251 576 | 11 176 512 |
| SCM | 11 786 488 | 609 976 | 11 176 512 |

**Table E.1. Amount of parameters in the different models**. There are many variations of the Concept-Shapes datasets, here we show the one resulting in the fewest parameters (top) and the most parameters (middle).

searched for the exponential learning rate decay parameter in $\{0.1, 0.7\}$, applied every ten epochs. The concept-based models set this 1 and searched for concept weight in $\{10, 15\}$.

The oracle models had a learning rate of 0.01 for ConceptShapes and 0.001 for CUB. They quickly converged and did not require further hyperparameter tuning. The grid search was implemented with the Python library Optuna [32].

We conducted the experiments on the University of Oslo's USIT ML nodes [33], where we used RTX 2080 Ti GPUs with 11GB VRAM. The hyperparameter search for the full CUB dataset, which consisted of 6 trials for 5 different models, took about 24 hours. For ConceptShapes, where we ran 48 trials for each of the 5 models, the hyperparameter search for the biggest subset configuration took about 12 hours.

# E   Training Details

We used the Adam optimizer [34] with a linear learning rate scheduler and the PyTorch deep learning library [35]. The standard model was constructed to be as similar as possible to the concept-based models. Instead of a bottleneck layer, it had an ordinary linear layer. We added dropout [36] after the convolutional part of the models. The amount of model parameters can be found in Table E.1. We used a cross entropy loss function as the class loss and a binary cross entropy loss function for the concepts.

The models trained on CUB used a frozen ResNet-18 [28] pre-trained on ImageNet [37] as the base model, while the ConceptShapes models were trained from scratch. We used three layers of $3 \times 3$ convolutional blocks, with padding and $2 \times 2$ max-pooling. We used 256 nodes in the first linear layer for the models trained on CUB, and 64 nodes for the models trained on ConceptShapes.

The pixel values were scaled down to $[0, 1]$ and normalized. The models trained on CUB used ImageNet [37] normalization parameters, and the models on the ConceptShapes datasets used means of 0.5 and standard deviations of 2 for all channels. We performed random cropping on the training images, and centre cropping when evaluating and testing. We did not perform any data augmentation that changed colours of the images, in order to not interfere with the concepts relating to colours.

