# OpenReview forum: "Hybrid Concept-based Models: Using Concepts to Improve Neural Networks’ Accuracy"
_NLDL.org/2026/Conference — NLDL 2026 Poster_

### Official Review · Reviewer_W8Xe · 2025-10-01
**Review of Manuscript**

**Rating:** 4
**Confidence:** 4

**Summary:**

This paper tackles the domain of learning from both images and concepts. The authors propose two methods of learning from both image concepts and labels. To evaluate such concept based learning they also propose a new benchmark called ConceptShape. Finally the also propose adversarial attacks where concepts are left unchanged but the classification label can change.

**Strengths:**

* The two proposed novel methods are sound. The experimental results highlight their strength.
* The intuition behind ConceptShape makes sense and it is developed to be interpretable .
* The authors were also able to show that such concept based methods are susceptible to adversarial attacks which bring into question their nature.

**Weaknesses:**

* Even though I see the intuition behind Concept Shape, I also think that for proper benchmarking some additional complexity is required.
* The experimental sections can benefit from more details about how the experiments were run for reproducability.

**Justification:**

This paper dives deep into the domain of concept based learning. The authors propose multiple improvements in this domain: better classification and benchmarking. Furthermore, the authors ask a fundamental question about the nature of concept based learning. Given these proposals, I am currently leaning towards an accept.

---

### Official Review · Reviewer_Ypkq · 2025-10-05
**Interesting approach to concept-based models with a nice dataset but too many independent ideas for one paper**

**Rating:** 2
**Confidence:** 4

**Summary:**

The paper introduces two new model architectures derived from the basic Concept Bottleneck Network (CBM) idea with a focus on improving performance of the model rather than having truthful explanations. The authors demonstrate, that leveraging concepts as an additional learning signal can indeed improve performance compared to classical CNNs and CBMs. The authors further introduce a new dataset with fine control over the number of concepts, their interrelation and their usefulness for predicting the final class. Finally, the authors investigate adversarial examples for CBMs proposing an algorithm that finds input samples that exhibit the same concepts as the original but with a different class output. Showing a high success rate of such attacks the authors question the notion of interpretability provided by such models.

**Strengths:**

Overall, I appreciate the idea of leveraging concept information to improve model performance rather that only focussing on interpretability. It provides an interesting angle, even though the range of applicable problems may be limited due to missing concept annotations for most datasets today. Nonetheless, the demonstration of a a clear benefit might lead to a shift from *more data samples* to *more information per sample* which is crucial in domains with limited samples and therefore highly relevant. Furthermore, the paper is well structured and written, it is therefore easy to follow and most of the concepts are clearly explained.

In my view, the main strengths of the paper are:

* **Model architectures:** Two different approaches of mixing concept information with a basic network architecture to combine the information value provided by both to increase model performance.
* **ConceptShapes dataset:** The proposed dataset is a valuable contribution to the research field in concept-based learning since availability of datasets with concept annotations is limited to begin with and a better control over the relationship of concepts and the target output is very much appreciated. It opens the door for other research, e.g. to investigate how good concept annotations have to be to provide any additional value.
* **Experimental results beyond performance:** In contrast to many other publications with a single focus on performance, the authors also investigate the usefulness (and correctness) of the predicted concepts and provide valuable insights in the relationship between concept predictions and final results.
* **Reproducibility:** Dataset and evaluation code are available.

**Weaknesses:**

In my opinion, the main weakness of the paper is that it tries to do too many things and once and therefore, misses the sufficient depth. While the investigation of the proposed model architectures and the description of the dataset are still ok, the adversarial topic would need much more work (and its own paper in my opinion).

Here are the main weaknesses I see:

* **Related work:** Strong focus on concept-based models. But a discussion of existing literature to the other two contributions (dataset, adversarial attacks) is mostly missing with the exception of one short sentence about the existence of attacks on concept predictions. This section could definitely improve also in order to further motivate the following propositions. In [1] the authors also demonstrate (as a side-effect) that additional information during training can improve model performance. [2] also introduces a dataset for concept learning based on shapes.
* **Adversarial Concept Attacks:** This topic feels rushed in the main paper. There is more information in the annex but compared to the other two aspects, it falls short in the main paper and the ideas are hard to comprehend without referring to the annex a lot. In my view this point it is also not necessary for the paper. I'd rather focus on the first two topics and extend the investigation/discussion for them.
Furthermore, I'm not completely convinced of the notion *concepts do not change*. In essence each concept is a binary classifier and *changing* refers to the fact that the output of this classifier goes from 1 to 0 (or vice versa) through some sort of threshold (usually 0.5?). However, CBMs use the scores (or even logits) a signal with much more information. Assume now that you have all concepts class to the threshold for changing - unless the following (linear) layer(s) is not perfectly aligned with all of those concepts, I don't find it surprising to find adversarial examples where lets say concept A is still slightly above the threshold, but the output of the overall model already changes because the classification layer afterwards already interprets it as "not there". I think, any investigation of this phenomenon should take the confidence in the individual concepts (i.e. the magnitude of the score) into account. Much further investigation is needed and would justify its own publication, I guess.
* **Ambiguous experimental setup:** The authors claim (in ll.294ff) that ResNet-18 (pre-trained and frozen) was used for the CUB dataset. However, I don't see how they easily incorporated their SCM in this architecture (especially when frozen) and how they selected which model was predicted at which layer (a problem that I don't see discussed anywhere btw). Why did the authors chose a frozen ResNet-18 without any fine-tuning during training? I'm not sure how well the pre-trained model represents the *bird details* relevant for the concepts. The model used for the ConceptShapes is hidden in annex E. From the main paper I would have inferred it used ResNet-18, too. Nonetheless, some of the design aspects seem a bit arbitrary and could benefit from further elaboration or ablation studies (other architectures, influence of pretraining etc.).
* **Colors in the plots:** Hard to distinguish different results, especially different types of green/yellow.
* **Short discussion of improved accuracy:** The discussion section is rather short and for the accuracy focusses solely on the ConceptShapes dataset and neglects the results on CUB. While I appreciate the comparison with the oracle model and the investigation of the influence of the concept information on the main performance, this section could be improved with CUB results and potential limitations of the conclusions that can be drawn from these experiments. Also, no comparison of the different proposed architectures is provided (beyond the plots itself).
* **CUB Concepts are not learned (ll. 341ff):** The authors themselves already give one reason, why the concepts are not learned properly (ambiguous and sometimes wrong annotations of concepts). I'd argue, that using a frozen backbone pre-trained on ImageNet may also contribute to this fact since the concepts for birds are highly specific and I doubt that the pre-trained model has sufficient features available to describe them properly. Therefore, this result is in part due to the (limited) choices for the experimental design (see one of the previous points). Therefore, I also question the authors claim that the different between CUB and ConceptShapes lies only in the ambiguity of the concepts but also in the hard to compare model architecture and learning schemes. Furthermore, concepts in the ConceptShapes dataset are most probably much easier to learn to begin with (and with less data). Therefore, I'm not entirely sure, what the contribution of section 5.2 really is.
* **Missing limitations:** Any discussion of limitations either for the proposed model architectures nor the ConceptShapes dataset is missing. The authors should critically reflect, what possible limitations of those approaches and the provided experimental results are. While I appreciate the overall idea, it feels like there are many open aspects (see discussion above) not investigated thoroughly enough. If it is not possible to address them in this paper, they should at least be noted as clear limitations.



[1] Schwaiger et al. (2023): "Preventing Errors in Person Detection: A Part-Based Self-Monitoring Framework", IEEE Intelligent Vehicle Symposium 2023

[2] Stammer et al. (2022): "Interactive Disentanglement: Learning Concepts by Interacting with their Prototype Representations", CVPR 2022

**Justification:**

The main idea of leveraging concept annotations as an additional training signal to improve performance as well as the proposed ConceptShapes dataset can be a valuable contribution, if the open questions (contribution of the base model architecture, comparison of the different proposed architecture enhancement, better disentanglement of pre-training and concept prediction quality, ...) are addressed properly. Many bits and pieces are already there and my initial feeling was towards *borderline / weak reject* - which I cannot select. In its current form, the paper tries to do too many things and I strongly recommend to give the whole *Adversarial Concept Attacks* branch more thought and its own paper, using the freed up space in here to address the open questions, provide more ablation studies on the questions above and investigate further why and how it works. I see potential in this work, but in my opinion it is not there yet.

---

### Official Review · Reviewer_qsqB · 2025-10-08

**Rating:** 1
**Confidence:** 5
**Final Rating:** 2
**Final Confidence:** 5

**Summary:**

This paper proposes hybrid concept-based models, new architectures designed to leverage both class labels and auxiliary concept information to improve model performance. Unlike traditional concept bottleneck models (CBMs) that focus on interpretability, the proposed models prioritize accuracy by allowing the network to use both concept predictions and non-concept features through skip connections.

**Strengths:**

The hybrid designs (CBM-Res, CBM-Skip, SCM) are simple yet effective extensions of existing CBMs.

**Weaknesses:**

I am very grateful to the authors for their contributions. However, the paper has some significant flaws:

* Unclear Motivation: While the authors highlight limitations in interpretability for existing models, the proposed approaches do not directly address this issue. Since the primary purpose of CBMs is interpretability rather than raw performance, there are numerous other strategies that could improve accuracy alone. The connection between the proposed models and improved interpretability is weak or unsubstantiated. Additionally, the paper claims that adversarial examples undermine interpretability, but this concern is not meaningfully resolved in the proposed methods.

* Interpretability Discussion is Superficial: While the paper critiques interpretability in concept-based models, it does not deeply analyze why or how hybrid models might mitigate or exacerbate interpretability loss.

* Limited Novelty in Architecture: The proposed hybrid designs (skip/residual and sequential prediction) are incremental modifications of standard CBMs rather than fundamentally new architectures.

* Limited Theoretical Insight: The paper does not provide a theoretical justification or formal analysis for why hybrid connections improve performance.

* Adversarial Evaluation Incomplete: The adversarial concept attacks are introduced but not thoroughly compared to strong baselines or analyzed in terms of perceptual similarity and robustness.

**Final Justification:**

I have read all the comments from the other reviewers as well as the authors’ response. I am not convinced by the authors’ reply, and I believe that this manuscript is not yet ready for publication.

While I acknowledge the value of empirical studies, as noted by reviewers qsqB and Ypkq, the experimental setup remains ambiguous, and the discussion of improved accuracy and interpretability is rather superficial. Consequently, it is difficult to clearly identify the contribution of this work.

**Justification:**

Due to the unclear motivation, limited architectural novelty, and insufficient analysis of interpretability and adversarial robustness, I recommend a strong rejection.

---

> ### Author Rebuttal · Authors · 2025-10-15
>
> Thank you very much for your valuable feedback. We would like to highlight some points of potential misunderstanding.
>
> ## Weaknesses
>
> ### Bulletpoint 1
>
> The motivation of the article is not to make interpretable models, but to improve the models’ performance. This is made clear in the introduction ll.052-ll.056, which states:, “*Due to the evidence demonstrating the limitations of interpretability in concept-based models, we will shift our focus away from interpretability and instead use the framework of concept-based models to improve the performance of the models.*”.
>
> Therefore, we do not claim that the proposed models are interpretable, but show empirical evidence that they achieve better performance than the CBM and CNN benchmark models. There are many approaches to improving accuracy, and in this paper we do it using more information per datapoint.
>
> The adversarial concept attacks adds to the evidence that concept-based models are not truly interpretable. This builds onto the reasons for the shift in motivation away from interpretability and over to performance. We do *not* claim that we resolve the issue of interpretability.
>
> ### Bulletpoint 2
>
> Since the motivation of this article is to improve performance rather than interpretability, we intentionally do not analyze the interpretability of the proposed models. Our goal is to demonstrate that concept-based architectures also can serve as a useful framework for improving predictive performance. Therefore, the discussion of interpretability is included only to motivate this shift in focus, not as an intended contribution of the work.
>
> ### Bulletpoint 3
>
> The proposed hybrid designs are indeed simple modifications of existing model architectures. However, the contribution lies in using the framework for concept-based models in order to improve performance rather than interpretability, and the addition of skip connections in various forms indeed confirms improved accuracy across datasets. The novelty therefore does not lie in the skip connections themselves, but how they are integrated in the existing framework.
>
> ### Bulletpoint 4
>
> We agree that a formal theoretical analysis is not provided. Our focus in this work is empirical, to demonstrate that the proposed hybrid concept-based architectures allow the models to better utilize both concept and non-concept information, resulting in improved predictive performance. Additionally, we contribute by providing a new concept dataset for benchmarking concept-based models and an algorithm for adversarial concept attacks, questioning the interpretable qualities of concept-based models. While a theoretical justification would be valuable, it is outside the scope of this study and is an interesting direction for future research.
>
> ### Bulletpoint 5
>
> The adversarial concept attacks works as a counterexample for the previous concept-based models’ promised interpretable qualities. The concept predictions from a CBM are used as the interpretation of its final prediction, so by altering the input so that the concept predictions change, but not the final prediction, this further adds on to the evidence of lack of interpretable qualities in CBMs. We do compare the adversarial concept attacks to a projected gradient descent (PGD) in Table 1. However, as the main argument of the adversarial concept attacks is to further question the interpretability qualities in CBMs and therefore change the focus away from interpretability and over to performance, an elaborate analysis of adversarial concept attacks is outside the scope of this article.

---

### Official Review · Reviewer_YWSc · 2025-10-09
**Good paper with relevant takeaway messages**

**Rating:** 5
**Confidence:** 4

**Summary:**

The authors present a broad undertaking of concept-based modeling for classification tasks, highlighting limitations in commonly used datasets while also pointing out caveats in relying upon concepts for explainability.  Shifting the focus to leverage additional concept labels to the benefit of the primary classification task, the authors propose two effective architectures improving upon state-of-the-art concept bottleneck models (CBMs).  Then, the authors propose a procedurally-generated dataset allowing control over the number of classes and concepts and the correlation thereof, and use it to analyze the models using both prediction accuracy and misprediction overlap.  Going back to the question of concept utility for explainability, the authors convincingly demonstrate successful adversarial attacks that alter concept predictions while leaving class predictions as is.

**Strengths:**

- Propose two architectures improving on the best known concept bottleneck models (CBMs), with adequate consideration of the utility of information in concept predictions.
- Propose a procedure to generate an image dataset with ground-truth labels and concepts, with a controllable level of concept-class correlation.
- Highlight critical caveats in relying upon concepts for explainable class predictions, by demonstrating successful adversarial attacks that alter concept predictions but not class predictions.

Suggestion: perhaps the title can be more compelling by mentioning the caveats exposed by the adversarial experiment, and advocates for a shift in focus to leverage concepts to improve accuracy rather than rely upon them for explainability.  Something along the lines of correlation but not causation could be worked in if the authors entertain it.

**Weaknesses:**

- Using a synthetic dataset of primitive shapes, allowing control over concept classes, is a good test bed and helps resolve some limitations of existing datasets as the authors briefly survey.  However, it makes for a more conceptual study though admittedly the findings are still relevant more broadly.
- It would be interesting to consider captioning/annotating more realistic image datasets - using the latest large models - to study a richer and more practically relevant setting.  Taking into account the inherent noise in such pseudo-labels for derived concepts would be interesting to study as well.

**Justification:**

The study and presentation are very well done, and present valuable takeaways for the ML community.

---

### Meta-Review · Area_Chair_nVR4 · 2025-11-01

**Recommendation:** Accept (Poster)
**Confidence:** 4

**Metareview:**

This work has several strengths (such as the investigation and repurposing of CBMs), but also has several weaknesses(including an unclear motivation and limited discussion). Overall, a good summary of this paper made by reviewer Ypkq was: "a bunch of good ideas but none of them really feels like they brought it to an end". Given this, I recommend **borderline** acceptance for this paper.

---

### Decision · Program_Chairs · 2025-11-05

**Decision:**

Accept (Poster)

**Comment:**

We recommend a poster presentation given the AC and reviewers recommendations.